# Go for a Walk and Arrive at the Answer: Reasoning Over Paths in Knowledge Bases using Reinforcement Learning

**Rajarshi Das**[*,1]**, Shehzaad Dhuliawala**[*,1]**, Manzil Zaheer**[*,2]
**Luke Vilnis**[1]**, Ishan Durugkar**[3]**, Akshay Krishnamurthy**[1]**, Alex Smola**[4]**, Andrew McCallum**[1]
`{rajarshi, sdhuliawala, luke, akshay, mccallum}@cs.umass.edu`
`manzil@cmu.edu, ishand@cs.utexas.edu, alex@smola.org`
[1]University of Massachusetts, Amherst, [2]Carnegie Mellon University
[3]University of Texas at Austin, [4]Amazon Web Services

## Abstract

Knowledge bases (KB), both automatically and manually constructed, are often incomplete — many valid facts can be inferred from the KB by synthesizing existing information. A popular approach to KB completion is to infer new relations by combinatory reasoning over the information found along other paths connecting a pair of entities. Given the enormous size of KBs and the exponential number of paths, previous path-based models have considered only the problem of predicting a missing relation given two entities, or evaluating the truth of a proposed triple. Additionally, these methods have traditionally used random paths between fixed entity pairs or more recently learned to pick paths between them. We propose a new algorithm, MINERVA, which addresses the much more difficult and practical task of answering questions where the relation is known, but only one entity. Since random walks are impractical in a setting with unknown destination and combinatorially many paths from a start node, we present a neural reinforcement learning approach which learns how to navigate the graph conditioned on the input query to find predictive paths. On a comprehensive evaluation on seven knowledge base datasets, we found MINERVA to be competitive with many current state-of-the-art methods.

## 1 Introduction

Automated reasoning, the ability of computing systems to make new inferences from observed evidence, has been a long-standing goal of artificial intelligence. We are interested in automated reasoning on large knowledge bases (KB) with rich and diverse semantics (Suchanek et al., 2007; Bollacker et al., 2008; Carlson et al., 2010). KBs are highly incomplete (Min et al., 2013), and facts not directly stored in a KB can often be inferred from those that are, creating exciting opportunities and challenges for automated reasoning. For example, consider the small knowledge graph in Figure 1. We can answer the question "Who did Malala Yousafzai share her Nobel Peace prize with?" from the following reasoning *path*: Malala Yousafzai → WonAward → Nobel Peace Prize 2014 → AwardedTo → Kailash Satyarthi. Our goal is to automatically learn such reasoning paths in KBs. We frame the learning problem as one of query answering, that is to say, answering questions of the form (Malala Yousafzai, SharesNobelPrizeWith, ?).

From its early days, the focus of automated reasoning approaches has been to build systems that can learn crisp symbolic logical rules (McCarthy, 1960; Nilsson, 1991). Symbolic representations have also been integrated with machine learning especially in statistical relational learning (Muggleton et al., 1992; Getoor & Taskar, 2007; Kok & Domingos, 2007; Lao et al., 2011), but due to poor generalization performance, these approaches have largely been superceded by distributed vector representations. Learning embedding of entities and relations using tensor factorization or neural methods has been a popular approach (Nickel et al., 2011; Bordes et al., 2013; Socher et al., 2013, inter alia), but these methods cannot capture chains of reasoning expressed by KB paths. Neural multi-hop models (Neelakantan et al., 2015; Guu et al., 2015; Toutanova et al., 2016) address the aforementioned problems to some extent by operating on KB paths embedded in vector space. However, these models take as input a set of paths which are gathered by performing random walks

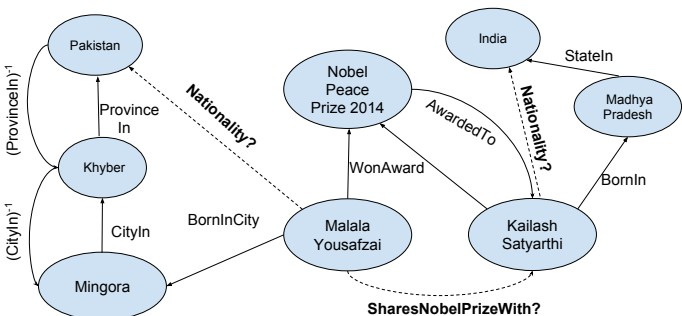

Figure 1: A small fragment of a knowledge base represented as a knowledge graph. Solid edges are observed and dashed edges are part of queries. Note how each query relation (e.g. SharesNobelPrizeWith, Nationality, etc.) can be answered by traversing the graph via "logical" paths between entity 'Malala Yousafzai' and the corresponding answer.

*independent* of the query relation. Additionally, models such as those developed in Neelakantan et al. (2015); Das et al. (2017) use the same set of initially collected paths to answer a diverse set of query types (e.g. MarriedTo, Nationality, WorksIn etc.).

This paper presents a method for efficiently searching the graph for answer-providing paths using reinforcement learning (RL) conditioned on the input question, eliminating any need for pre-computed paths. Given a massive knowledge graph, we learn a policy, which, given the query $(entity_1, relation, ?)$, starts from $entity_1$ and learns to walk to the answer node by choosing to take a labeled relation edge at each step, *conditioning* on the query relation and entire path history. This formulates the query-answering task as a reinforcement learning (RL) problem where the goal is to take an optimal sequence of decisions (choices of relation edges) to maximize the expected reward (reaching the correct answer node). We call the RL agent MINERVA for "Meandering In Networks of Entities to Reach Verisimilar Answers."

Our RL-based formulation has many desirable properties. First, MINERVA has the built-in flexibility to take paths of variable length, which is important for answering harder questions that require complex chains of reasoning (Shen et al., 2017). Secondly, MINERVA needs *no pretraining* and trains on the knowledge graph from scratch with reinforcement learning; no other supervision or fine-tuning is required representing a significant advance over prior applications of RL in NLP. Third, our path-based approach is computationally efficient, since by searching in a small neighborhood around the query entity it avoids ranking all entities in the KB as in prior work. Finally, the reasoning paths found by our agent automatically form an interpretable provenance for its predictions.

The main contributions of the paper are: (a) We present agent MINERVA, which learns to do query answering by walking on a knowledge graph conditioned on an input query, stopping when it reaches the answer node. The agent is trained using reinforcement learning, specifically policy gradients (§ 2). (b) We evaluate MINERVA on several benchmark datasets and compare favorably to Neural Theorem Provers (NTP) (Rocktäschel & Riedel, 2017) and Neural LP (Yang et al., 2017), which do logical rule learning in KBs, and also state-of-the-art embedding based methods such as DistMult (Yang et al., 2015) and ComplEx (Trouillon et al., 2016) and ConvE (Dettmers et al., 2018). (c) We also extend MINERVA to handle partially structured natural language queries and test it on the WikiMovies dataset (§ 3.3) (Miller et al., 2016).

We also compare to DeepPath (Xiong et al., 2017) which uses reinforcement learning to pick paths between entity pairs. The main difference is that the state of their RL agent includes the answer entity since it is designed for the simpler task of predicting if a fact is true or not. As such their method cannot be applied directly to our more challenging query answering task where the second entity is unknown and must be inferred. Nevertheless, MINERVA outperforms DeepPath on their benchmark NELL-995 dataset when compared in their experimental setting (§ 3.2.2).

## 2    TASK AND MODEL

We formally define the task of query answering in a KB. Let $\mathcal{E}$ denote the set of entities and $\mathcal{R}$ denote the set of binary relations. A KB is a collection of facts stored as triplets $(e_1, r, e_2)$ where $e_1, e_2 \in \mathcal{E}$ and $r \in \mathcal{R}$. From the KB, a knowledge graph $\mathcal{G}$ can be constructed where the entities $e_1, e_2$ are represented as the nodes and relation $r$ as labeled edge between them. Formally, a knowledge graph is a directed labeled multigraph $\mathcal{G} = (V, E, \mathcal{R})$, where $V$ and $E$ denote the vertices and edges of the graph respectively. Note that $V = \mathcal{E}$ and $E \subseteq V \times \mathcal{R} \times V$. Also, following previous approaches

(Bordes et al., 2013; Neelakantan et al., 2015; Xiong et al., 2017), we add the inverse relation of every edge, i.e. for an edge $(e_1, r, e_2) \in E$, we add the edge $(e_2, r^{-1}, e_1)$ to the graph. (If the set of binary relations $\mathcal{R}$ does not contain the inverse relation $r^{-1}$, it is added to $\mathcal{R}$ as well.)

Since KBs have a lot of missing information, two natural tasks have emerged in the information extraction community - fact prediction and query answering. Query answering seeks to answer questions of the form $(e_1, r, ?)$, e.g. Toronto, locatedIn, ?, whereas fact prediction involves predicting if a fact is true or not, e.g. (Toronto, locatedIn, Canada)?. Algorithms for fact prediction can be used for query answering, but with significant computation overhead, since all candidate answer entities must be evaluated, making it prohibitively expensive for large KBs with millions of entities. In this work, we present a query answering model, that learns to efficiently traverse the knowledge graph to find the correct answer to a query, eliminating the need to evaluate all entities.

Query answering reduces naturally to a finite horizon sequential decision making problem as follows: We begin by representing the environment as a deterministic partially observed Markov decision process on a knowledge graph $\mathcal{G}$ derived from the KB (§2.1). Our RL agent is given an input query of the form $(e_{1q}, r_q, ?)$. Starting from vertex corresponding to $e_{1q}$ in $\mathcal{G}$, the agent follows a path in the graph stopping at a node that it predicts as the answer (§ 2.2). Using a training set of known facts, we train the agent using policy gradients more specifically by REINFORCE (Williams, 1992) with control variates (§ 2.3). Let us begin by describing the environment.

## 2.1 Environment - States, Actions, Transitions and Rewards

Our environment is a finite horizon, deterministic partially observed Markov decision process that lies on the knowledge graph $\mathcal{G}$ derived from the KB. On this graph we will now specify a deterministic partially observed Markov decision process, which is a 5-tuple $(\mathcal{S}, \mathcal{O}, \mathcal{A}, \delta, R)$, each of which we elaborate below.

**States**. The state space $\mathcal{S}$ consists of all valid combinations in $\mathcal{E} \times \mathcal{E} \times \mathcal{R} \times \mathcal{E}$. Intuitively, we want a state to encode the query $(e_{1q}, r_q)$, the answer $(e_{2q})$, and a location of exploration $e_t$ (current location of the RL agent). Thus overall a state $S \in \mathcal{S}$ is represented by $S = (e_t, e_{1q}, r_q, e_{2q})$ and the state space consists of all valid combinations.

**Observations**. The complete state of the environment is not observed. Intuitively, the agent knows its current location $(e_t)$ and $(e_{1q}, r_q)$, but not the answer $(e_{2q})$, which remains hidden. Formally, the observation function $\mathcal{O} : \mathcal{S} \rightarrow \mathcal{E} \times \mathcal{E} \times \mathcal{R}$ is defined as $\mathcal{O}(s = (e_t, e_{1q}, r_q, e_{2q})) = (e_t, e_{1q}, r_q)$.

**Actions**. The set of possible actions $\mathcal{A}_S$ from a state $S = (e_t, e_{1q}, r_q, e_{2q})$ consists of all outgoing edges of the vertex $e_t$ in $\mathcal{G}$. Formally $\mathcal{A}_S = \{(e_t, r, v) \in E : S = (e_t, e_{1q}, r_q, e_{2q}), r \in \mathcal{R}, v \in V\} \cup \{(s, \varnothing, s)\}$. Basically, this means an agent at each state has option to select which outgoing edge it wishes to take having the knowledge of the label of the edge $r$ and destination vertex $v$.

During implementation, we unroll the computation graph up to a fixed number of time steps T. We augment each node with a special action called 'NO_OP' which goes from a node to itself. Some questions are easier to answer and needs fewer steps of reasoning than others. This design decision allows the agent to remain at a node for any number of time steps. This is especially helpful when the agent has managed to reach a correct answer at a time step $t < T$ and can continue to stay at the 'answer node' for the rest of the time steps. Alternatively, we could have allowed the agent to take a special 'STOP' action, but we found the current setup to work sufficiently well. As mentioned before, we also add the inverse relation of a triple, i.e. for the triple $(e_1, r, e_2)$, we add the triple $(e_2, r^{-1}, e_1)$ to the graph. We found this important because this actually allows our agent to undo a potentially wrong decision.

**Transition**. The environment evolves deterministically by just updating the state to the new vertex incident to the edge selected by the agent. The query and answer remains the same. Formally, the transition function is $\delta : \mathcal{S} \times \mathcal{A} \rightarrow \mathcal{S}$ defined by $\delta(S, A) = (v, e_{1q}, r_q, e_{2q})$, where $S = (e_t, e_{1q}, r_q, e_{2q})$ and $A = (e_t, r, v)$).

**Rewards**. We only have a terminal reward of +1 if the current location is the correct answer at the end and 0 otherwise. To elaborate, if $S_T = (e_t, e_{1q}, r_q, e_{2q})$ is the final state, then we receive a reward of +1 if $e_t = e_{2q}$ else 0., i.e. $R(S_T) = \mathbb{I}\{e_t = e_{2q}\}$.

## 2.2 POLICY NETWORK

To solve the finite horizon deterministic partially observable Markov decision process described above, we design a randomized non-stationary history-dependent policy $\pi = (\mathbf{d_1}, \mathbf{d_2}, ..., \mathbf{d_{T-1}})$, where $\mathbf{d_t} : H_t \to \mathcal{P}(\mathcal{A}_{S_t})$ and history $H_t = (H_{t-1}, A_{t-1}, O_t)$ is just the sequence of observations and actions taken. We restrict ourselves to policies parameterized by long short-term memory network (LSTM) (Hochreiter & Schmidhuber, 1997).

An agent based on LSTM encodes the history $H_t$ as a continuous vector $\mathbf{h_t} \in \mathbb{R}^{2d}$. We also have embedding matrix $\mathbf{r} \in \mathbb{R}^{|\mathcal{R}| \times d}$ and $\mathbf{e} \in \mathbb{R}^{|\mathcal{E}| \times d}$ for the binary relations and entities respectively. The history embedding for $H_t = (H_{t-1}, A_{t-1}, O_t)$ is updated according to LSTM dynamics:

$$\mathbf{h_t} = \text{LSTM}\left(\mathbf{h_{t-1}}, [\mathbf{a_{t-1}}; \mathbf{o_t}]\right) \tag{1}$$

where $\mathbf{a_{t-1}} \in \mathbb{R}^d$ and $\mathbf{o_t} \in \mathbb{R}^d$ denote the vector representation for action/relation at time $t-1$ and observation/entity at time $t$ respectively and $[;]$ denote vector concatenation. To elucidate, $\mathbf{a_{t-1}} = \mathbf{r}_{A_{t-1}}$, i.e. the embedding of the relation corresponding to label of the edge the agent chose at time $t-1$ and $\mathbf{o_t} = \mathbf{e}_{e_t}$ if $O_t = (e_t, e_{1q}, r_q)$ i.e. the embedding of the entity corresponding to vertex the agent is at time $t$.

Based on the history embedding $\mathbf{h_t}$, the policy network makes the decision to choose an action from all available actions $(\mathcal{A}_{S_t})$ *conditioned* on the query relation. Recall that each possible action represents an outgoing edge with information of the edge relation label $l$ and destination vertex/entity $d$. So embedding for each $A \in \mathcal{A}_{S_t}$ is $[\mathbf{r_l}; \mathbf{e_d}]$, and stacking embeddings for all the outgoing edges we obtain the matrix $\mathbf{A_t}$. The network taking these as inputs is parameterized as a two-layer feed-forward network with ReLU nonlinearity which takes in the current history representation $\mathbf{h_t}$ and the embedding for the query relation $\mathbf{r_q}$ and outputs a probability distribution over the possible actions from which a discrete action is sampled. In other words,

$$\mathbf{d_t} = \text{softmax}\left(\mathbf{A_t}(\mathbf{W_2}\text{ReLU}\left(\mathbf{W_1}\left[\mathbf{h_t}; \mathbf{o_t}; \mathbf{r_q}\right]\right))\right),$$
$$A_t \sim \text{Categorical}\left(\mathbf{d_t}\right).$$

Note that the nodes in $\mathcal{G}$ do not have a fixed ordering or number of edges coming out from them. The size of matrix $\mathbf{A_t}$ is $|\mathcal{A}_{S_t}| \times 2d$, so the decision probabilities $d_t$ lies on simplex of size $|\mathcal{A}_{S_t}|$. Also the procedure above is invariant to order in which edges are presented as desired and falls in purview of neural networks designed to be permutation invariant (Zaheer et al., 2017). Finally, to summarize, the parameters of the LSTM, the weights $\mathbf{W_1}$, $\mathbf{W_2}$, the corresponding biases (not shown above for brevity), and the embedding matrices form the parameters $\theta$ of the policy network.

## 2.3 TRAINING

For the policy network ($\pi_\theta$) described above, we want to find parameters $\theta$ that maximize the expected reward:

$$J(\theta) = \mathbb{E}_{(e_1, r, e_2) \sim D} \mathbb{E}_{A_1, ..., A_{T-1} \sim \pi_\theta}[R(S_T)|S_1 = (e_1, e_1, r, e_2)],$$

where we assume there is a true underlying distribution $(e_1, r, e_2) \sim D$. To solve this optimization problem, we employ REINFORCE (Williams, 1992) as follows:

- The first expectation is replaced with empirical average over the training dataset.

- For the second expectation, we approximate by running multiple rollouts for each training example. The number of rollouts is fixed and for all our experiments we set this number to 20.

- For variance reduction, a common strategy is to use an additive control variate baseline (Hammersley, 2013; Fishman, 2013; Evans & Swartz, 2000). We use a moving average of the cumulative discounted reward as the baseline. We tune the weight of this moving average as a hyperparameter. Note that in our experiments we found that using a learned baseline performed similarly, but we finally settled for cumulative discounted reward as the baseline owing to its simplicity.

- To encourage diversity in the paths sampled by the policy at training time, we add an entropy regularization term to our cost function scaled by a constant ($\beta$).

| Dataset | #entities | #relations | #facts | #queries | #degree | |
|---------|-----------|------------|--------|----------|---------|---|
| | | | | | avg. | median |
| COUNTRIES | 272 | 2 | 1158 | 24 | 4.35 | 4 |
| UMLS | 135 | 49 | 5,216 | 661 | 38.63 | 28 |
| KINSHIP | 104 | 26 | 10686 | 1074 | 82.15 | 82 |
| WN18RR | 40,945 | 11 | 86,835 | 3134 | 2.19 | 2 |
| NELL-995 | 75,492 | 200 | 154,213 | 3992 | 4.07 | 1 |
| FB15K-237 | 14,505 | 237 | 272,115 | 20,466 | 19.74 | 14 |
| WikiMovies | 43,230 | 9 | 196,453 | 9952 | 6.65 | 4 |

Table 1: Statistics of various datasets used in experiments.

| | ComplEx | ConvE | DistMult | NTP | NTP-$\lambda$ | NeuralLP | MINERVA |
|---|---------|-------|----------|-----|--------------|----------|---------|
| S1 | 99.37±0.4 | **100.0±0.00** | 97.91±0.01 | 90.83±15.4 | **100.0±0.00** | **100.0±0.0** | **100.0±0.00** |
| S2 | 87.95±2.8 | **99.0±1.00** | 69.18±2.38 | 87.40±11.7 | 93.04±0.40 | 75.1 ± 0.3 | 92.36±2.41 |
| S3 | 48.44±6.3 | 86.0±5.00 | 15.79±0.64 | 56.68±17.6 | 77.26±17.0 | 92.2 ± 0.2 | **95.10±1.20** |

Table 2: Performance on three tasks of COUNTRIES dataset with AUC-PR metric. MINERVA significantly outperforms all other methods on the hardest task (S3). Also variance across runs for MINERVA is lower compared to other methods.

## 3 EXPERIMENTS

We now present empirical studies for MINERVA in order to establish that (i) MINERVA is competitive for query answering on small (Sec. 3.1.1) as well as large KBs (Sec. 3.1.2), (ii) MINERVA is superior to a path based models that do not search the KB efficiently or train query specific models (Sec. 3.2), (iii) MINERVA can not only be used for well formed queries, but can also easily handle partially structured natural language queries (Sec 3.3), (iv) MINERVA is highly capable of reasoning over long chains, and (v) MINERVA is robust to train and has much faster inference time (Sec. 3.5).

### 3.1 KNOWLEDGE BASE QUERY ANSWERING

To gauge the reasoning capability of MINERVA, we begin with task of query answering on KB, i.e. we want to answer queries of the form $(e_1, r, ?)$. Note that, as mentioned in Sec. 2, this task is subtly different from fact checking in a KB. Also, as most of the previous literature works in the regime of fact checking, their ranking includes variations of both $(e_1, r, x)$ and $(x, r, e_2)$. However, since we do not have access to $e_2$ in case of question answering scenario the same ranking procedure does not hold for us – we only need to rank on $(e_1, r, x)$. This difference in ranking made it necessary for us to re-run all the implementations of previous work. We used the implementation or the best pre-trained models (whenever available) of Rocktäschel & Riedel (2017); Yang et al. (2017) and Dettmers et al. (2018). For MINERVA to produce a ranking of answer entities during inference, we do a beam search with a beam width of 50 and rank entities by the probability of the trajectory the model took to reach the entity and remaining entities are given a rank of $\infty$.

**Method** We compare MINERVA with various state-of-the-art models using HITS@1,3,10 and mean reciprocal rank (MRR), which are standard metrics for KB completion tasks. In particular we compare against embedding based models - DistMult (Yang et al., 2015), ComplEx (Trouillon et al., 2016) and ConvE (Dettmers et al., 2018). For ConvE and ComplEx, we used the implementation released by Dettmers et al. (2018)[1] on the best hyperparameter settings reported by them. For DistMult, we use our highly tuned implementation (e.g. which performs better than the state-of-the-art results of Toutanova et al. (2015)). We also compare with two recent work in learning logical rules in KB namely Neural Theorem Provers (NTP) (Rocktäschel & Riedel, 2017) and NeuralLP (Yang et al., 2017). Rocktäschel & Riedel (2017) also reports a NTP model which is trained with an additional objective function of ComplEx (NTP-$\lambda$). For these models, we used the implementation released by corresponding authors [2] [3], again on the best hyperparameter settings reported by them.

---

[1] https://github.com/TimDettmers/ConvE

[2] https://github.com/uclmr/ntp

[3] https://github.com/fanyangxyz/Neural-LP

| Data | Metric | ComplEx | ConvE | DistMult | NTP | NTP-λ | NeuralLP | MINERVA |
|------|--------|---------|-------|----------|-----|-------|----------|---------|
| KINSHIP | HITS@1 | 0.754 | 0.697 | 0.808 | 0.500 | 0.759 | 0.475 | 0.605 |
| | HITS@3 | 0.910 | 0.886 | 0.942 | 0.700 | 0.798 | 0.707 | 0.812 |
| | HITS@10 | 0.980 | 0.974 | 0.979 | 0.777 | 0.878 | 0.912 | 0.924 |
| | MRR | 0.838 | 0.797 | 0.878 | 0.612 | 0.793 | 0.619 | 0.720 |
| UMLS | HITS@1 | 0.823 | 0.894 | 0.916 | 0.817 | 0.843 | 0.643 | 0.728 |
| | HITS@3 | 0.962 | 0.964 | 0.967 | 0.906 | 0.983 | 0.869 | 0.900 |
| | HITS@10 | 0.995 | 0.992 | 0.992 | 0.970 | 1.000 | 0.962 | 0.968 |
| | MRR | 0.894 | 0.933 | 0.944 | 0.872 | 0.912 | 0.778 | 0.825 |

Table 3: Query answering results on KINSHIP and UMLS datasets.

### 3.1.1 SMALLER DATASETS

**Dataset** We use three standard datasets: COUNTRIES (Bouchard et al., 2015), KINSHIP, and UMLS (Kok & Domingos, 2007). The COUNTRIES dataset ontains countries, regions, and subregions as entities and is carefully designed to explicitly test the logical rule learning and reasoning capabilities of link prediction models. The queries are of the form LocatedIn(c, ?) and the answer is a region (e.g. LocatedIn(Egypt, ?) with the answer as Africa). The dataset has 3 tasks (S1-3 in table 2) each requiring reasoning steps of increasing length and difficulty (see Rocktäschel & Riedel (2017) for more details about the tasks). Following the design of the COUNTRIES dataset, for task S1 and S2, we set the maximum path length $T = 2$ and for S3, we set $T = 3$. The Unified Medical Language System (UMLS) dataset, is from biomedicine. The entities are biomedical concepts (e.g. disease, antibiotic) and relations are like treats and diagnoses. The KINSHIP dataset contains kinship relationships among members of the Alyawarra tribe from Central Australia. For these two task we use maximum path length $T = 2$. Also, for MINERVA we turn off entity in (1) in these experiments.

**Observations** For the COUNTRIES dataset, in Table 2 we report a stronger metric - the area under the precision-recall curve - as is common in the literature. We can see that MINERVA compares favorably or outperforms all the baseline models except on the task S2 of COUNTRIES, where the ensemble model NTP-λ and ConvE outperforms it, albeit with a higher variance across runs. Our gains are much more prominent in task S3, which is the hardest among all the tasks.

The Kinship and UMLS datasets are small KB datasets with around 100 entities each and as we see from Table 3, embedding based methods (ConvE, ComplEx and DistMult) perform much better than methods which aim to learn logical rules (NTP, NeuralLP and MINERVA). On Kinship, MINERVA outperforms both NeuralLP and NTP and matches the HITS@10 performance of NTP on UMLS. Unlike COUNTRIES, these datasets were not designed to test the logical rule learning ability of models and given the small size, embedding based models are able to get really high performance. Combination of both methods gives a slight increase in performance as can be seen from the results of NTP-λ. However, when we initialized MINERVA with pre-trained embeddings of ComplEx, we did not find a significant increase in performance.

### 3.1.2 LARGER DATASETS

**Dataset** Next we evaluate MINERVA on three large KG datasets - WN18RR, FB15K-237 and NELL-995. The WN18RR (Dettmers et al., 2018) and FB15K-237 (Toutanova et al., 2015) datasets are created from the original WN18 and FB15K datasets respectively by removing various sources of test leakage, making the datasets more realistic and challenging. The NELL-995 dataset released by Xiong et al. (2017) has separate graphs for each query relation, where a graph for a query relation can have triples from the test set of another query relation. For the query answering experiment, we combine all the graphs and removed all test triples (and the corresponding triples with inverse relations) from the graph. We also noticed that several triples in the test set had an entity (source or target) that never appeared in the graph. Since, there will be no trained embeddings for those entities, we removed them from the test set. This reduced the size of test set from 3992 queries to 2818 queries.[4]

---

[4]Available at `https://github.com/shehzaadzd/MINERVA`

| Data | Metric | ComplEx | ConvE | DistMult | NeuralLP | Path-Baseline | MINERVA |
|------|--------|---------|-------|----------|----------|---------------|---------|
| WN18RR | HITS@1 | 0.382 | 0.403 | 0.410 | 0.376 | 0.017 | 0.413 |
|  | HITS@3 | 0.433 | 0.452 | 0.441 | 0.468 | 0.025 | 0.456 |
|  | HITS@10 | 0.480 | 0.519 | 0.475 | 0.657 | 0.046 | 0.513 |
|  | MRR | 0.415 | 0.438 | 0.433 | 0.463 | 0.027 | 0.448 |
| FB15K-237 | HITS@1 | 0.303 | 0.313 | 0.275 | 0.166 | 0.169 | 0.217 |
|  | HITS@3 | 0.434 | 0.457 | 0.417 | 0.248 | 0.248 | 0.329 |
|  | HITS@10 | 0.572 | 0.600 | 0.568 | 0.348 | 0.357 | 0.456 |
|  | MRR | 0.394 | 0.410 | 0.370 | 0.227 | 0.227 | 0.293 |
| NELL-995 | HITS@1 | 0.612 | 0.672 | 0.610 | - | 0.300 | 0.663 |
|  | HITS@3 | 0.761 | 0.808 | 0.733 | - | 0.417 | 0.773 |
|  | HITS@10 | 0.827 | 0.864 | 0.795 | - | 0.497 | 0.831 |
|  | MRR | 0.694 | 0.747 | 0.680 | - | 0.371 | 0.725 |

Table 4: Query answering results on WN18RR, FB15K-237 and NELL-995 datasets. NeuralLP does not scale to NELL-995 and hence the entries are kept blank.

**Observations**   Table 4 reports the query answering results on the larger WN18RR, FB15K-237 and NELL-995 datasets. We could not include the results of NeuralLP on NELL-995 since it didn't scale to that size. Similarly NTP did not scale to any of the larger datasets. Apart from these, we are the first to report a comprehensive summary of performance of all baseline methods on these datasets.

On NELL-995, MINERVA performs comparably to embedding based methods such as DistMult and ComplEx and performs comparably with ConvE on the stricter HITS@1 metric. ConvE, however outperforms us on HITS@10 on NELL-995. On WN18RR, logic based based methods (NeuralLP, MINERVA) generally outperform embedding based methods, with MINERVA achieving the highest score on HITS@1 metric and NeuralLP significantly outperforming on HITS@10.

We observe that on FB15K-237, however, embedding based methods dominate over MINERVA and NeuralLP. Upon deeper inspection, we found that the query relation types of FB15K-237 knowledge graph differs significantly from others.

Analysis of query relations of FB15k-237: We analyzed the type of query relation types on the FB15K-237 dataset. Following Bordes et al. (2013), we categorized the query relations into (M)any to 1, 1 to M or 1 to 1 relations. An example of a M to 1 relation would be '/people/profession' (What is the profession of person 'X'?). An example of 1 to M relation would be /music/instrument/instrumentalists ('Who plays the music instrument X?') or '/people/ethnicity/people' ('Who are people with ethnicity X?'). From a query answering point of view, the answer to these questions is a list of entities. However, during evaluation time, the model is evaluated based on whether it is able to predict the one target entity which is in the query triple. Also, since MINERVA outputs the end points of the paths as target entities, it is sometimes possible that the particular target entity of the triple does not have a path from the source entity (however there are paths to other 'correct' answer entities). Table 9 (in appendix) shows few other examples of relations belonging to different classes.

Following Bordes et al. (2013), we classify a relation as 1-to-M if the ratio of cardinality of tail to head entities is greater than 1.5 and as M-to-1 if it is lesser than 0.67. In the validation set of FB15K-237, 54% of the queries are 1-to-M, whereas only 26% are M-to-1. Contrasting it with NELL-995, 27% are 1-to-M and 36% are M-to-1 or UMLS where only 18% are 1-to-M. Table 10 (in appendix) shows few relations from FB15K-237 dataset which have high tail-to-head ratio. The average ratio for 1-TO-M relations in FB15K-237 is **13.39** (substantially higher than 1.5). As explained before, the current evaluation scheme is not suited when it comes to 1-to-M relations and the high percentage of 1-to-M relations in FB15K-237 also explains the sub optimal performance of MINERVA.

We also check the frequency of occurrence of various unique path types. We define a path type as the sequence of relation types (ignoring the entities) in a path. Intuitively, a predictive path which generalizes across queries will occur many number of times in the graph. Figure 2 shows the plot. As we can see, the characteristics of FB15K-237 is quite different from other datasets. For example, in NELL-995, more than 1000 different path types occur more than 1000 times. WN18RR has only 11 different relation types which means there are only $11^3$ possible path types of length 3 and even fewer number of them would be predictive. As can be seen, there are few path types which occur more than $10^4$ times and around 50 of them occur more than 1000 times. However in FB15K-237,

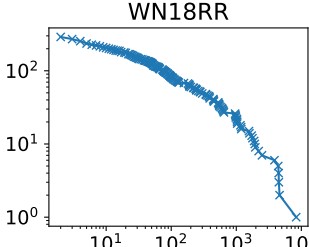 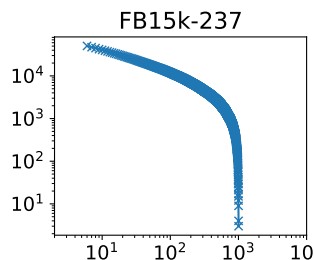 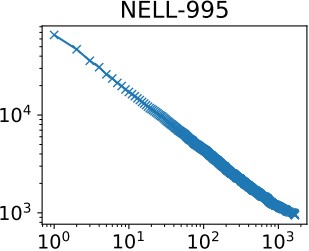

Figure 2: Count of number of unique path types of length 3 which occur more than 'x' times in various datasets. For example, in NELL-995 there are more than $10^3$ path types which occur more than $10^3$ times. However, for FB15k-237, we see a sharp decrease as 'x' becomes higher, suggesting that path types do not repeat often.

which has the highest number of relation types, we observe a sharp decrease in the number of path types which occur a significant number of times. Since MINERVA cannot find path types which repeat often, it finds it hard to learn path types that generalize.

## 3.2 COMPARISON WITH PATH BASED MODELS

### 3.2.1 WITH RANDOM WALK MODELS

In this experiment, we compare to a model which gathers path based on random walks and tries to predict the answer entity. Neural multi-hop models (Neelakantan et al., 2015; Toutanova et al., 2016), operate on paths between entity pairs in a KB. However these methods need to know the target entity in order to pre-compute paths between entity pairs. (Guu et al., 2015) is an exception in this regard as they do random walks starting from a source entity '$e_1$' and then using the path, they train a classifier to predict the target answer entity. However, they only consider *one* path starting from a source entity. In contrast, Neelakantan et al. (2015); Toutanova et al. (2016) use information from multiple paths between the source and target entity. We design a baseline model which combines the strength of both these approaches. Starting from '$e_1$', the model samples ($k = 100$) random paths of up to a maximum length of $T = 3$. Following Neelakantan et al. (2015), we encode each paths with an LSTM followed by a max-pooling operation to featurize the paths. This feature is concatenated with the source entity and query relation vector which is then passed through a feed forward network which scores all possible target entities. The network is trained with a multi-class cross entropy objective based on observed triples and during inference we rank target entities according to the model score.

The PATH-BASELINE column of table 4 shows the performance of this model on the three datasets. As we can see MINERVA outperforms this baseline significantly. This shows that a model which predicts based on a set of randomly sampled paths does not do as well as MINERVA because it either loses important paths during random walking or it fails to aggregate predictive features from all the $k$ paths, many of which would be irrelevant to answer the given query. The latter is akin to the problem with distant supervision (Mintz et al., 2009), where important evidence gets lost amidst a plethora of irrelevant information. However, by taking each step conditioned on the query relation, MINERVA can effectively reduce the search space and focus on paths relevant to answer the query.

### 3.2.2 WITH DEEPPATH

We also compare MINERVA with DeepPath which uses RL to pick paths between entity pairs. For a fair comparison, we only rank the answer entities against the negative examples in the dataset used in their experiments[5] and report the mean average precision (MAP) scores for each query relation. DeepPath feeds the paths its agent gathers as input features to the path ranking algorithm (PRA) (Lao et al., 2011), which trains a per-relation classifier. But unlike them, we train one model which learns for all query relations so as to enable our agent to leverage from correlations and more data. If our agent is not able to reach the correct entity or one of the negative entities, the corresponding entities gets a score of negative infinity. If MINERVA fails to reach any of the entities in the set of correct and negative entities. then we fall back to a random ordering of the entities. As show in

---

[5]We are grateful to Xiong et al. (2017) for releasing the negative examples used in their experiments.

| Task | DeepPath | MINERVA | MINERVA[a] |
|---|---|---|---|
| athleteplaysinleague | 0.960 | **0.970** | 0.940 |
| worksfor | 0.711 | **0.825** | 0.810 |
| organizationhiredperson | 0.742 | **0.851** | 0.856 |
| athleteplayssport | 0.957 | **0.985** | 0.980 |
| teamplayssport | 0.738 | **0.846** | 0.880 |
| personborninlocation | 0.757 | **0.793** | 0.780 |
| personleadsorganization | 0.795 | **0.851** | 0.877 |
| athletehomestadium | 0.890 | **0.895** | 0.898 |
| organizationheadquarteredincity | 0.790 | **0.946** | 0.940 |
| athleteplaysforteam | 0.750 | **0.824** | 0.800 |

Table 5: MAP scores for different query relations on the NELL-995 dataset. Note that in this comparison, MINERVA refers to only a single learnt model for all query relations which is competitive with individual DeepPath models trained separately for each query relation. We also trained MINERVA in the setting of DeepPath, i.e. training per-relation models (MINERVA[a])

table 5, we outperform them or achieve comparable performance for all the query relations For this experiment, we set the maximum length $T = 3$. Although training per-relation models is cumbersome and does not scale to massive KBs with thousands of relation types, we also train per-relation models of MINERVA replicating the settings of DeepPath (MINERVA[a] in table 5). MINERVA[a] outperforms DeepPath and performs similarly to MINERVA which is an encouraging result since training one model which performs well for all relation is highly desirable.

## 3.3 PARTIALLY STRUCTURED QUERIES

Queries in KBs are structured in the form of triples. However, this is unsatisfactory since for most real applications, the queries appear in natural language. As a first step in this direction, we extend MINERVA to take in "partially structured" queries. We use the WikiMovies dataset (Miller et al., 2016) which contains questions in natural language albeit generated by templates created by human annotators. An example question is "Which is a film written by Herb Freed?". WikiMovies also has an accompanying KB which can be used to answer all the questions.

| Model | Accuracy |
|---|---|
| Memory Network | 78.5 |
| QA system | 93.5 |
| Key-Value Memory Network | 93.9 |
| Neural LP | 94.6 |
| MINERVA | **96.7** |

Table 6: Performance on WikiMovies

We link the entity occurring in the question to the KB via simple string matching. To form the vector representation of the query relation, we design a simple question encoder which computes the average of the embeddings of the question words. The word embeddings are learned from scratch and we do not use any pretrained embeddings. We compare our results with those reported in Yang et al. (2017) (table 6). For this experiment, we found that $T = 1$ sufficed, suggesting that WikiMovies is not the best testbed for multihop reasoning, but this experiment is a promising first step towards the realistic setup of using KBs to answer natural language question.

## 3.4 GRID WORLD PATH FINDING

While chains in KB need not be very long to get good empirical results (Neelakantan et al., 2015; Das et al., 2017; Yang et al., 2017), in principle MINERVA can be used to learn long reasoning chains. To evaluate the same, we test our model on a synthetic 16-by-16 grid world dataset created by Yang et al. (2017), where the task is to navigate to a particular cell (answer entity) starting from a random cell (start entity) by following a set of directions (query relation). The KB consists of atomic triples of the form ((2,1), North, (1,1)) – entity (1,1) is north of entity (2,1). The queries consists of a sequence of directions (e.g. North, SouthWest, East). The queries are classified into classes based on the path lengths. Figure 3 shows the accuracy on varying path lengths. Compared to Neural LP, MINERVA is much more robust to queries, which require longer path, showing minimal degradation in performance for even the longest path in the dataset.

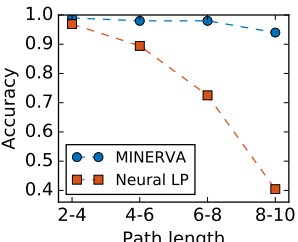

Figure 3: Grid world experiment: We significantly outperform NeuralLP for longer path lengths.

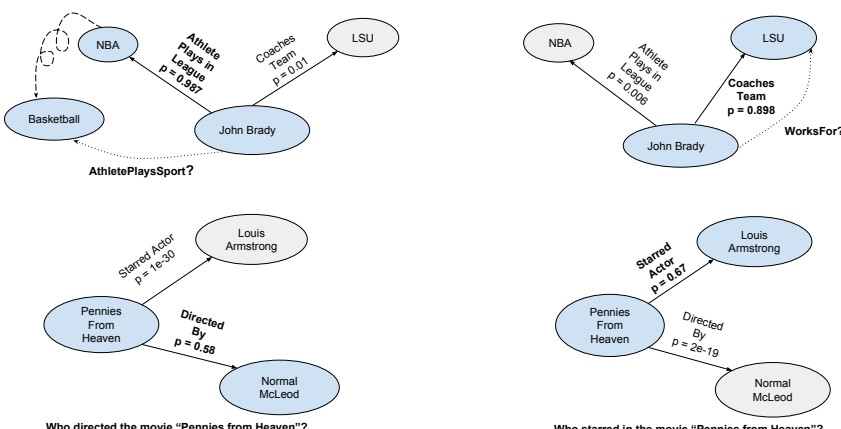

Figure 4: Based on the query relation our agent assigns different probabilities to different actions. The dashed edges in the top row denote query relation. Examples in the bottom row are from the WikiMovies dataset and hence the questions are partially structured.

## 3.5 FURTHER ANALYSIS

**Training time.** Figure 5 plots the HITS@10 scores on the development set against the training time comparing MINERVA with DistMult. It can be seen that MINERVA converges to a higher score much faster than DistMult. It is also interesting to note that even during the early stages of the training, MINERVA has much higher performance than that of DistMult, as during these initial stages, MINERVA would just be doing random walks in the neighborhood of the source entity ($e_1$). This implies that MINERVA's approach of searching for an answer in the neighborhood of $e_1$ is a much more efficient and smarter strategy than ranking all entities in the knowledge graph (as done by DistMult and other related methods).

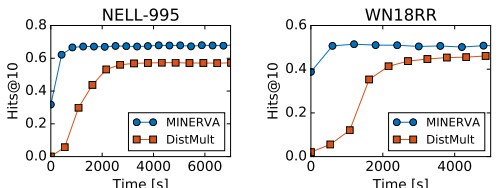

Figure 5: HITS@10 on the development set versus training time.

**Inference Time.** At test time, embedding based methods such as ConvE, ComplEx and DistMult rank all entities in the graph. Hence, for a test-time query, the running time is always $\mathcal{O}(|\mathcal{E}|)$ where $\mathcal{R}$ denotes the set of entities (= nodes) in the graph. MINERVA, on the other hand is efficient at inference time since it has to essentially search for answer entities in its local neighborhood. The many cost at inference time for MINERVA is to compute probabilities for all outgoing edges along the path. Thus inference time of MINERVA only depends on degree distribution of the graph. If we assume the knowledge graph to obey a power law degree distribution, like many natural graphs, then for MINERVA the average inference time can be shown to be $O(\frac{\alpha}{\alpha-1})$, when the coefficient of the power law $\alpha > 1$. The median inference time for MINERVA is $O(1)$ for all values of $\alpha$. Note that these quantities are independent of size of entities $|\mathcal{E}|$. For instance, on the test dataset of WN18RR, the wall clock inference time of MINERVA is **63s** whereas that of a GPU implementation of DistMult, which is the simplest among the lot, is **211s**. Similarly the wall-clock inference time on the test set of NELL-995 for a GPU implementation of DistMult is 115s whereas that of MINERVA is **35s**.

**Query based Decision Making.** At each step before making a decision, our agent conditions on the query relation. Figure 4 shows examples, where based on the query relation, the probabilities are peaked on different actions. For example, when the query relation is WorksFor, MINERVA assigns a much higher probability of taking the edge CoachesTeam than AthletePlaysInLeague. We also see similar behavior on the WikiMovies dataset where the query consists of words instead of fixed schema relation.

**Model Robustness.** Table 7 also reports the mean and standard deviation across three independent runs of MINERVA. We found it easy to obtain/reproduce the highest scores across several runs as can be seen from the low deviations in scores.

| Dataset | HITS@1 | HITS@3 | HITS@10 |
|---------|--------|--------|---------|
| NELL-995 | $0.66 \pm 0.029$ | $0.77 \pm 0.0016$ | $0.83 \pm 0.0030$ |
| FB15K-237 | $0.22 \pm 0.002$ | $0.33 \pm 0.0008$ | $0.46 \pm 0.0006$ |
| WN18RR | $0.41 \pm 0.030$ | $0.45 \pm 0.0180$ | $0.51 \pm 0.0005$ |

Table 7: Mean and Standard deviation across runs for various datasets.

**Effectiveness of Remembering Path History.** MINERVA encodes the history of decisions it has taken in the past using LSTMs. To test the importance of remembering the sequence of decisions, we did an ablation study in which the agent chose the next action based on only local information i.e. current entity and query and did not have access to the history $h_t$. For the KINSHIP dataset, we observe a 27% points decrease in HITS@1 and 13% decrease in HITS@10. For grid-world, it is also not surprising that we see a big drop in performance. The final accuracy is 0.23 for path lengths 2-4 and 0.04 for lengths 8-10. For FB15K-237 the HITS@10 performance dropped from 0.456 to 0.408.

**NO-OP and Inverse Relations.** At each step, MINERVA can choose to take a NO-OP edge and remain at the same node. This gives the agent the flexibility of taking paths of variable lengths. Some questions are easier to answer than others and require fewer steps of reasoning and if the agent reaches the answer early, it can choose to remain there. Example (i) in table 8 shows such an example. Similarly inverse relation gives the agent the ability to recover from a potentially wrong decision it has taken before. Example (ii) shows such an example, where the agent took a incorrect decision at the first step but was able to revert the decision because of the presence of inverted edges.

## 4 RELATED WORK

Learning vector representations of entities and relations using tensor factorization (Nickel et al., 2011; 2012; Bordes et al., 2013; Riedel et al., 2013; Nickel et al., 2014; Yang et al., 2015) or neural methods (Socher et al., 2013; Toutanova et al., 2015; Verga et al., 2016) has been a popular approach to reasoning with a knowledge base. However, these methods cannot capture more complex reasoning patterns such as those found by following inference paths in KBs. Multi-hop link prediction approaches (Lao et al., 2011; Neelakantan et al., 2015; Guu et al., 2015; Toutanova et al., 2016; Das et al., 2017) address the problems above, but the reasoning paths that they operate on are gathered by performing random walks independent of the type of query relation. Lao et al. (2011) further filters paths from the set of sampled paths based on the restriction that the path must end at one of the target entities in the training set and are within a maximum length. These constraints make them query

---

(i) **Can learn general rules:**

(S1) LocatedIn(X, Y) ← LocatedIn(X, Z) & LocatedIn(Z, Y)
(S2) LocatedIn(X, Y) ← NeighborOf(X, Z) & LocatedIn(Z, Y)
(S3) LocatedIn(X, Y) ← NeighborOf(X, Z) & NeighborOf(Z, W) & LocatedIn(W, Y)

---

(ii) **Can learn shorter path:** Richard F. Velky $\xrightarrow{\text{WorksFor}}$ ?

Richard F. Velky $\xrightarrow{\text{PersonLeadsOrg}}$ Schaghticokes $\xrightarrow{\text{NO-OP}}$ Schaghticokes $\xrightarrow{\text{NO-OP}}$ Schaghticokes

---

(iii) **Can recover from mistakes:** Donald Graham $\xrightarrow{\text{WorksFor}}$ ?

Donald Graham $\xrightarrow{\text{OrgTerminatedPerson}}$ TNT Post $\xrightarrow{\text{OrgTerminatedPerson}^{-1}}$ Donald Graham $\xrightarrow{\text{OrgHiredPerson}}$ Wash Post

---

Table 8: A few example of paths found by MINERVA on the COUNTRIES and NELL. MINERVA can learn general rules as required by the COUNTRIES dataset (example (i)). It can learn shorter paths if necessary (example (ii)) and has the ability to correct a previously taken decision (example (iii))
.

dependent but they are heuristic in nature. Our approach eliminates any necessity to pre-compute paths and learns to efficiently search the graph conditioned on the input query relation.

Inductive Logic Programming (ILP) (Muggleton et al., 1992) aims to learn general purpose predicate rules from examples and background knowledge. Early work in ILP such as FOIL (Quinlan, 1990), PROGOL (Muggleton, 1995) are either rule-based or require negative examples which is often hard to find in KBs (by design, KBs store true facts). Statistical relational learning methods (Getoor & Taskar, 2007; Kok & Domingos, 2007; Schoenmackers et al., 2010) along with probabilistic logic (Richardson & Domingos, 2006; Broecheler et al., 2010; Wang et al., 2013) combine machine learning and logic but these approaches operate on symbols rather than vectors and hence do not enjoy the generalization properties of embedding based approaches.

There are few prior work which treat inference as search over the space of natural language. Nogueira & Cho (2016) propose a task (WikiNav) in which each the nodes in the graph are Wikipedia pages and the edges are hyperlinks to other wiki pages. The entity is to be represented by the text in the page and hence the agent is required to reason over natural language space to navigate through the graph. Similar to WikiNav is Wikispeedia (West et al., 2009) in which an agent needs to learn to traverse to a given target entity node (wiki page) as quickly as possible. Angeli & Manning (2014) propose natural logic inference in which they cast the inference as a search from a query to any valid premise. At each step, the actions are one of the seven lexical relations introduced by MacCartney & Manning (2007).

Neural Theorem Provers (NTP) (Rocktäschel & Riedel, 2017) and Neural LP (Yang et al., 2017) are methods to learn logical rules that can be trained end-to-end with gradient based learning. NTPs are constructed by Prolog's backward chaining inference method. It operates on vectors rather than symbols, thereby providing a success score for each proof path. However, since a score can be computed between any two vectors, the computation graph becomes quite large because of such *soft-matching* during substitution step of backward chaining. For tractability, it resorts to heuristics such as only keeping the top-K scoring proof paths trading-off guarantees for exact gradients. Also the efficacy of NTPs has yet to be shown on large KBs. Neural LP introduces a differential rule learning system using operators defined in TensorLog (Cohen, 2016). It has a LSTM based controller with a differentiable memory component (Graves et al., 2014; Sukhbaatar et al., 2015) and the rule scores are calculated via attention. Even though, differentiable memory allows end to end training, it necessitates accessing the entire memory, which can be computationally expensive. RL approaches capable of hard selection of memory (Zaremba & Sutskever, 2015) are computationally attractive. MINERVA uses a similar hard selection of relation edges to walk on the graph. More importantly, MINERVA outperforms both these methods on their respective benchmark datasets.

DeepPath (Xiong et al., 2017) uses RL based approaches to find paths in KBs. However, the state of their MDP requires the target entity to be known in advance and hence their path finding strategy is dependent on knowing the answer entity. MINERVA does not need any knowledge of the target entity and instead learns to find the answer entity among all entities. DeepPath, additionally feeds its gathered paths to Path Ranking Algorithm (Lao et al., 2011), whereas MINERVA is a complete system trained to do query answering. DeepPath also uses fixed pretrained embeddings for its entity and relations. Lastly, on comparing MINERVA with DeepPath in their experimental setting on the NELL dataset, we match their performance or outperform them. MINERVA is also similar to methods for learning to search for structured prediction (Collins & Roark, 2004; Daumé III & Marcu, 2005; Daumé III et al., 2009; Ross et al., 2011; Chang et al., 2015). These methods are based on imitating a reference policy (oracle) which make near-optimal decision at every step. In our problem setting, it is unclear what a good reference policy would be. For example, a shortest path oracle between two entities would be unideal, since the answer providing path should depend on the query relation.

## 5 CONCLUSION

We explored a new way of automated reasoning on large knowledge bases in which we use the knowledge graphs representation of the knowledge base and train an agent to walk to the answer node conditioned on the input query. We achieve state-of-the-art results on multiple benchmark knowledge base completion tasks and we also show that our model is robust and can learn long chains-of-reasoning. Moreover it needs no pretraining or initial supervision. Future research directions include applying more sophisticated RL techniques and working directly on textual queries and documents.

ACKNOWLEDGEMENTS

We are grateful to Patrick Verga for letting us use his implementation of DistMult. This work was supported in part by the Center for Data Science and the Center for Intelligent Information Retrieval, in part by DARPA under agreement number FA8750-13-2-0020, in part by Defense Advanced Research Agency (DARPA) contract number HR0011-15-2-0036, in part by the National Science Foundation (NSF) grant numbers DMR-1534431 and IIS-1514053 and in part by the Chan Zuckerberg Initiative under the project Scientific Knowledge Base Construction. The U.S. Government is authorized to reproduce and distribute reprints for Governmental purposes notwithstanding any copyright notation thereon. Any opinions, findings and conclusions or recommendations expressed in this material are those of the authors and do not necessarily reflect those of the sponsor.

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

---

(i) **M to 1**

Los Angeles Rams $\xrightarrow{\text{team plays sport}}$ American Football

The Walking Dead $\xrightarrow{\text{country of origin}}$ USA

---

(ii) **1 to M**

CEO $\xrightarrow{\text{job position in organization}}$ Merck & Co.

Traffic collision $\xrightarrow{\text{cause of death}}$ Albert Camus

Harmonica $\xrightarrow{\text{instrument played by musician}}$ Greg Graffin

---

Table 9: Few example facts belonging to m to 1, 1 to m relations in FB15K-237
.

| Relation | tail/head |
|---|---|
| /people/marriage_union_type/unions_of_this_type./people/marriage/location_of_ceremony | 129.75 |
| /organization/role/leaders./organization/leadership/organization | 65.15 |
| /location/country/second_level_divisions | 49.18 |
| /user/ktrueman/default_domain/international_organization/member_states | 36.5 |
| /base/marchmadness/ncaa_basketball_tournament/seeds./base/marchmadness/ncaa_tournament_seed/team | 33.6 |

Table 10: Few example 1-to-M relations from FB15K-237 with high cardinality ratio of tail to head.

# 6 APPENDIX

## 6.1 HYPERPARAMETERS

**Experimental Details** We choose the relation and embedding dimension size as 200. The action embedding is formed by concatenating the entity and relation embedding. We use a 3 layer LSTM with hidden size of 400. The hidden layer size of MLP (weights $\mathbf{W_1}$ and $\mathbf{W_2}$) is set to 400. We use Adam (Kingma & Ba, 2014) with the default parameters in REINFORCE for the update.

In our experiments, we tune our model over two hyper parameters, *viz.,* $\beta$ which is the entropy regularization constant and $\lambda$ which is the moving average constant for the REINFORCE baseline. The table 11 lists the best hyper parameters for all the datasets.

| Dataset | $\beta$ | $\lambda$ | Path Length |
|---|---|---|---|
| UMLS | 0.05 | 0.05 | 2 |
| KINSHIP | 0.1 | 0.05 | 2 |
| Countries S1 | 0.01 | 0.1 | 2 |
| Countries S2 | 0.02 | 0.1 | 2 |
| Countries S3 | 0.01 | 0.1 | 3 |
| WN18RR | 0.05 | 0.05 | 3 |
| NELL-995 | 0.06 | 0.0 | 3 |
| FB15K-237 | 0.02 | 0.05 | 3 |
| WIKIMOVIES | 0.15 | 0 | 1 |

Table 11: Best hyper parameters

## 6.2 ADDENDUM TO NELL RESULTS

The NELL dataset released by Xiong et al. (2017) includes two additional tasks for which the scores were not reported in the paper and so we were unable to compare them against DeepPath. Nevertheless, we ran MINERVA on these tasks and report our results in table 12 for completeness.

| Task | Single Model | DeepPath setup |
|---|---|---|
| agentbelongstoorganization | 0.86 | 0.87 |
| teamplaysinleague | 0.97 | 0.95 |

Table 12: NELL results for the remaining tasks

