# OpenReview forum: "Go for a Walk and Arrive at the Answer: Reasoning Over Paths in Knowledge Bases using Reinforcement Learning"
_ICLR.cc/2018/Conference — Accept (Poster)_

### Official Review · AnonReviewer1 · 2017-11-29

**Rating:** 6
**Confidence:** 4

**Review:**

The paper proposes an approach for query answering/link prediction in KBs that uses RL to navigate the KB graph between a query entity and a potential answer entity. The main originality is that, unlike random walk models, the proposed approach learns to navigate the graph while being conditioned on the query relation type.

I find the method sound and efficient and the proposed experiments are solid and convincing; for what they test for.

Indeed, for each relation type that one wants to be testing on, this type of approach needs many training examples of pairs of entities (say e_1, e_2) connected both by this relation type (e_1 R e_2) and by alternative paths (e_1 R' R'' R''' e_2). Because the model needs to discover and learn that R <=> R ' R'' R''' .

The proposed model seems to be able to do that well when the number of relation types remains low (< 50). But things get interesting in KBs when the number of relation types gets pretty large (hundreds / thousands). Learning the kind of patterns described above gets much trickier then. The results on FB15k are a bit worrying in that respect. Maybe this is a matter of the dataset FB15k itself but then having experiments on another dataset with hundreds of relation types could be important.

NELL has indeed 200 relations but if I'm not mistaken, the NELL dataset is used for fact prediction and not query answering. And as noted in the paper, fact prediction is much easier.

---

> ### Author Response · Authors · 2017-12-13
> **Thank you for your helpful reviews -- Paper updated with new experiments!**
>
> Thank you for your helpful reviews!
>
> You raised an interesting point regarding the performance of MINERVA on KGs with large number of relation types. For a fair comparison, we ran query answering (not fact prediction) experiments on NELL-995 and compared to our implementation of DistMult (which does very well on FB15k-237). DistMult achieves a score of 79.5 whereas MINERVA achieves a score of 82.73. Another important point to note is that MINERVA is much more efficient at inference time. NELL has ~75k entities and algorithms such as DistMult have to rank against all entities to get the final score. However MINERVA just has to walk to the right answer. This can be seen by comparing the wall-clock running times 35 secs wrt 115 secs - (sec 4.1 of the paper Query Answering on NELL-995)
> This empirically shows that MINERVA works for relations with many relation types. We would additionally like to point out that MINERVA does well on WikiMovies. In WikiMovies the queries are partially structured and are in natural language. Hence the number of query types are actually quite large (and potentially unbounded). This also supports our claim.  Thanks for the excellent suggestion again.
>
> We also have updated the paper with a detailed analysis of the negative results on Fb15k-237 (sec 4.1) and more importantly, how this dataset differs from other KG datasets.

---

### Official Review · AnonReviewer3 · 2017-11-30
**Simple RL approach for reasoning on Knowledge bases, good performance on variety of datasets but need to be slightly more thorough in their comparisons with prior work**

**Rating:** 7
**Confidence:** 4

**Review:**

The paper present a RL based approach to walk on a knowledge graph to answer queries. The idea is novel, the paper is clear in its exposition, and the authors provide a number of experimental comparisons with prior work on a variety of datasets .

Pros:
1. The approach is simple (no pre-training, no reward shaping, just RL from scratch with terminal reward, uses LSTM for keeping track of past state), computationally efficient (no computation over the full graph), and performs well in most of the experiments reported in the paper.
2. It scales well to longer path lengths, and also outperforms other methods for partially structured queries.

Cons:
1. You should elaborate more on the negative results on FB15K and why this performance would not transfer to other KB datasets that exist. This seems especially important since it's a large scale dataset, while the datasets a)-c) reported in the paper are small scale.
2. It would also be good to see if your method also performed well on the Nations dataset where the baselines performed well. That said, if its a small scale dataset, it would be preferable to focus on strengthening the experimental analysis on larger datasets.
3. In Section 4.2, why have you only compared to NeuralLP and not compared with the other methods?

Suggestions/Questions:
1. In the datatset statistics, can you also add the average degree of the knowledge graphs, to get a rough sense of the difficulty of each task.
2. The explanation of the knowledge graph and notation could be made cleaner. It would be easier to introduce the vertices as the entities, and edges as normal edges with a labelled relation on top. A quick example to explain the action space would also help.
3. Did you try a model where instead of using A_t directly as the weight vector for the softmax, you use it as an extra input? Using it as the weight matrix directly might be over regularizing/constraining your model.

Revision: I appreciate the effort by the authors to update the paper. All my concerns were adequately addressed, plus improvements were made to better understand the comparison with other work. I update my review to 7: Good paper, accept.

---

> ### Author Response · Authors · 2017-12-13
> **Thank you for your helpful reviews**
>
> Thank you for your helpful reviews. We have updated the paper with your suggestions.
> - We have updated the paper with a detailed analysis of the negative results on Fb15k-237 (sec 4.1) and more importantly, how this dataset differs from other KG datasets. (To summarize, FB15k-237 has very (i) low clustering coefficient (ii) the path types don't repeat that often (iii) has a lot of 1-Many query relations.)
>
> - In section 4.2 (Grid World), we actually compared to other baselines such as DistMult, but since they are not path based method their performance was very low. We decided to not report the results because it was making the plot look disproportionate. However for completion, here are the numbers - (Path length 2-4) 0.2365, (4-6) 0.1144, (6-8) 0.0808, (8-10)  0.0413
>
> - We have added the avg and mean degree of nodes of knowledge graphs in table 1. MINERVA performs well in KGs with both high/low out degree of nodes.
>
> - Yes!, we did consider using A_t as apart of the input but it comes with few complications w.r.t the implementation. First as the number and ordering of outgoing edges from a node varies, feeding A_t into the MLP is not straightforward. Also since the output probabilities should have support only on the outgoing edges (which are not uniquely determined by only the relations, but also the neighboring entity), the masking logic also becomes tricky. Finally, the excessive amount of parameters required this way might lead to overfitting. Since we were getting promising results with the simpler approach, we decided to continue with the first design choice.

---

> ### Author Response · Authors · 2018-01-15
> **Thank you!**
>
> We appreciate your revision of scores and we thank you for your helpful reviews once again!

---

### Official Review · AnonReviewer2 · 2017-11-30

**Rating:** 5
**Confidence:** 4

**Review:**

The paper proposes a new approach (Minerva) to perform query answering on knowledge bases via reinforcement learning. The method is intended to answer queries of the form (e,r,?) on knowledge graphs consisting of dyadic relations. Minerva is evaluated on a number of different datasets such as WN18, NELL-995, and WikiMovies.

The paper proposes interesting ideas to attack a challenging problem, i.e., how to perform query answering on incomplete knowledge bases. While RL methods for KG completion have been proposed recently (e.g., DeepPath), Minerva improves over these approaches by not requiring the target entity. This property can be indeed be important to perform query answering efficiently. The proposed model seems technically reasonable and the paper is generally written well and good to understand. However, important parts of the paper seem currently unfinished and would benefit from a more detailed discussion and analysis.

Most importantly, I'm currently missing a better motivation and especially a more thorough evaluation on how Minerva improves over non-RL methods. For instance, the authors mention multi-hop methods such as (Neelakantan, 2015; Guu, 2015) in the introduction. Since these methods are closely related, it would be important to compare to them experimentally (unfortunately, DeepPath doesn't do this comparison either). For instance, eliminating the need to pre-compute paths might be irrelevant when it doesn't improve actual performance. Similarly, the paper mentions improved inference time, which indeed is a nice feature. However, I'm wondering, what is the training time and how does it compare to standard methods like ComplEx. Also, how robust is training using REINFORCE?

With regard to the experimental results: The  improvements over DeepPath on NELL and on WikiMovies are indeed promising. I found the later results the most convincing, as the setting is closest to the actual task of query answering. However, what is worrying is that Minerva doesn't do well on WN18 and FB15k-237 (for which the results are, unfortunately, only reported in the appendix). On FB15k-237 (which is harder than WN18 and arguably more relevant for real-world scenarios since it is a subset of a real-world knowledge graph), it is actually outperformed by the relatively simple DistMult method. From these results, I find it hard to justify that "MINERVA obtains state-of-the-art results on seven KB datasets, significantly outperforming prior methods", as stated in the abstract.

Further comments:
- How are non-existing relations handled, i.e., queries (e,r,x) where there is no valid x? Does Minerva assume there is always a valid answer?
- Comparison to DeepPath: Did you evaluate Minerva with fixed embeddings? Since the experiments in DeepPath used fixed embeddings, it would be important to know how much of the improvements can be attributed to this difference.
- The experimental section covers quite a lot of different tasks and datasets (Countries, UMLS, Nations, NELL, WN18RR, Gridworld, WikiMovies) all with different combinations of methods. For instance, countries is evaluated against ComplEx,NeuralLP and NTP; NELL against DeepPath; WN18RR against ConvE, ComplEx, and DistMult; WikiMovies against MemoryNetworks, QA and NeuralLP. A more focused evaluation with a consistent set of methods could make the experiments more insightful.

---

> ### Author Response · Authors · 2018-01-05
> **Paper updated with additional experiments**
>
> Thank you for your helpful reviews!
>
> a) Comparison with multi-hop models: Thanks to your suggestion, we have updated the paper (sec 4.2) with a new experiment which explicitly compares to non-RL neural multihop models which precomputes a set of paths. Starting from the source entity, the model featurizes a set of paths (using a LSTM) and max-pools across them. This feature vector is then concatenated with the query relation and then fed to a feed forward network to score each target entity. This model is similar to that of Neelakantan et al. (2015) except for the fact that it was originally designed to work between a fixed set of source and target entity pair, but in our case the target entity is unknown. The model (baseline) is trained with a multi-class cross entropy objective based on observed triples and during inference we rank target entities according the the score given by the model.
> As we can see, MINERVA outperforms this model in both freebase and NELL suggesting that RL based approach can effectively reduce the search space and focus on paths relevant to answer the query. Please see sec 4.2 for additional details and results.
>
> b) Training time and robustness of the model:- We actually found MINERVA to be very robust during training. We were able to achieve the reported results without much tuning and they are also very easy to reproduce. However to quantify the results, we report the variance of the results across three independent runs on the freebase and nell datasets. Also we report the learning curve of score on the development set wrt time. (Please see sec 4.5 of the paper)
>
> c) Regarding peformance on WN18RR and FB15k-237:
> MINERVA actually achieves state-of-the-art in the WN18RR dataset.
> On FB15k-237, MINERVA matches with all baseline model and is outperformed only by DistMult. We have updated the paper with a detailed analysis of the negative results on Fb15k-237 (sec 4.1) and more importantly, how this dataset differs from other KG datasets. (To summarize, FB15k-237 has very (i) low clustering coefficient (ii) the path types don't repeat that often (iii) has a lot of 1-Many query relations.)
>
> d) Query Answering experiment on NELL-995: - We also added a query answering (not fact prediction) experiment on NELL-995 and compared to our implementation of DistMult (which does very well on FB15k-237). DistMult achieves a score of 79.5 whereas MINERVA achieves a score of 82.73. Another important point to note is that MINERVA is much more efficient at inference time. NELL has ~75k entities and algorithms such as DistMult have to rank against all entities to get the final score. However MINERVA just has to walk to the right answer. This can be seen by comparing the wall-clock running times 35 secs wrt 115 secs - (sec 4.1 of the paper Query Answering on NELL-995)
>
> Further  comments:
>
> a) How are non-existing relations handled, i.e., queries (e,r,x) where there is no valid x? Does Minerva assume there is always a valid answer? - That is a good point. Currently MINERVA does not support non existing relations and assumes there is always a valid answer. The ability to handle non-existing relations is definitely important and we plan to incorporate this in future work.
>
> b) Comparison to DeepPath: Did you evaluate Minerva with fixed embeddings? Since the experiments in DeepPath used fixed embeddings, it would be important to know how much of the improvements can be attributed to this difference
> We actually tried both cases - train randomly initialized embeddings from scratch and using fixed pretrained embeddings. We achieved similar results in both cases. For fixed embeddings, the model converged faster but to a similar score. However for uniformity across experiments, we reported results where we trained the embeddings.
>
> c) Consistent baselines: We will update the paper to cover as many reported baselines as possible. However we have made sure to the best of our abilities to compare with the models which have current state of the art results on each dataset.

---

### Public Comment · (anonymous) · 2017-11-05
**Negative results on FB15k-237**

Kudos for reporting negative results. Quick question: the results for the other methods on FB15k-237 are obtained by running code or copying results from previous work? I’m skeptical of/surprised by the 56.8 hits@10 for DistMult.

---

> ### Author Response · Authors · 2017-11-08
> **Re: Negative results on FB15k-237**
>
> Thank you for the comment. The performance of DistMult on FB15k-237 (56.8 HITS@10) are scores that we got with our own in-house implementation. To the best of our knowledge, it is better than all of the published scores for DistMult on this dataset (closest being 52.93 by Jain et al (2017) https://arxiv.org/pdf/1706.00637.pdf and 52.3 by Toutanova et al. (2015) http://cs.stanford.edu/~danqi/papers/emnlp2015.pdf). The results of ConvE and ComplEx models were taken from Dettmers et al. (2017) https://arxiv.org/pdf/1707.01476.pdf). We are aware of the high variance in scores reported for the DistMult model on FB15k-237 by many other papers, but we decided to report the results that we got for it. We will make this clear in the next version of the paper. Thanks again!

---

### Public Comment · (anonymous) · 2017-11-12
**Supervised approach**

A simple solution for this problem is supervised approach, i.e., treating the path from the source node to target node as *sequences* and training a RNN on these *sequences*. I am wondering how would the proposed approaches compare with the supervised version.

Intuitively, the reinforced version over supervised version is its efficiency in getting positive rewards. But for this problem, enumerating the paths from the source nodes to target nodes sounds like a more efficient way.

---

> ### Author Response · Authors · 2017-11-14
> **Re: Supervised Approach**
>
> Thank you for the comment. Since we are doing query answering, during test time, we do not have the information about target entities. That would mean enumerating all paths starting from a source entity and training a classifier to choose one of the target entities that these paths end on. This seemed like a reasonable first step, but the main bottleneck is the huge number of paths that need to be considered. For instance, the avg. number of length 3 paths starting from an entity in the validation set of Fb15k-237 is 1,800,759 (1.8M). That means during inference, we need to gather around 1.8M paths for each query, compute features and choose from one of the end entities. Sub-sampling paths is another approach, but it is difficult to come up with a non-heuristic way of subsampling. Another drawback of this approach that we would like to point out, is that only a few paths are ‘predictive’ of a query relation and subsampling might easily lose them.
>
> Yet another way of training using supervised approach would be to sample one path which leads to the target entity and another path which doesn’t and do a gradient update to favor the path which does reach. During inference time, we just sample from the model (RNN for example) a path and return the endpoint as the answer. This approach differs from our RL based approach in that during training we depend on the current model to sample the next path. This has the advantage of utilizing the information that the model has acquired by exploring the graphs till now. For example, it might have learned that a particular kind of path isn’t good for a query relation, even though it might lead to the target entity for this particular query. The RL approach will use that information and not select that path and would instead try to search a path which would generalize more.
>
> In general, it is a good idea to explore around where the model is, instead of doing uniform sampling and has been well studied in contextual bandit settings (Dudik et al., 2011, Agarwal et al., 2014). While the proposal is not doing uniform exploration since it samples any correct path to the target, even this kind of "uniform" sampling can hurt generalization performance since it does not use the current model to help identify which good paths are representable. Thanks again!
>
> Efficient Optimal Learning for Contextual Bandits - Dudik et al., 2011
> Taming the Monster: A Fast and Simple Algorithm for Contextual Bandits - Agarwal et al., 2014

---

> ### Author Response · Authors · 2018-01-05
> **Re: Supervised experiment**
>
> We have updated the paper with a new experiment which explicitly compares with a neural multi-hop (path) model which is designed to work in the query answering setting (when the target entity is unknown). Please refer to sec 4.2 of the paper. Thanks!

---

### Author Response · Authors · 2017-11-13
**Minor clarifications**

We wanted to address/clarify few minor mistakes which we found in the text of the paper which could possibly confuse the reviewers. We will definitely fix these in the next version of the paper.

1. Introduction: We suggest that PRA (Lao et al., 2011) use the same set of collected path to answer diverse query types. PRA, given a query, keeps paths which are supported by at least a fraction of the training queries and are also bounded by a maximum length. Additionally, they are constrained to end at one of the target entities in the training set. These constraints make them query dependent but are heuristic in nature which is in contrast to MINERVA which learns to use the query relation to make decisions at every step.
2. Sec 2.1 - Our environment is a finite horizon deterministic Markov decision model -> This should be finite horizon, deterministic and partially observed Markov decision process.
3. Sec 2.1 - From the KB, a knowledge graph G can be constructed where the entities s,t are represented as the nodes -> the identifiers 's' and 't' should be 'e1' and 'e2’.
4. Section 5 (Effectiveness of Remembering Path History) - We replaced the LSTM with a simple 2 layer MLP -> ‘replace’ might be confusing. In this ablation study, the policy network (2 layer MLP) makes decision based on only the local information (current entity, query). It does not have access to the entire history of decisions encoded by the LSTM.
5. Figure 3 - Caption - The figure in the right -> The figure at the bottom. Also for consistency, the top figure (left) should have LSU grayed and figure (right) should have NBA grayed.
6. Figure 1 - The edge from nodes USA to CA is wrongly labeled as 'has_city'. It should be instead labeled as 'has_state'.
7.  Figure 4 - This plot shows the frequency of occurrence of various unique paths (types) of length 3 which occur more than 'x' times in various datasets. Intuitively, a predictive path which generalizes across queries will occur many number of times in the graph. As we can see, the characteristics of FB15k-237 is quite different from other datasets. Path types do not repeat that often, making it hard for MINERVA to learn paths which generalizes.

---

### Public Comment · (anonymous) · 2017-11-13
**Biased Results Presentation - negative results should be included in result section instead of appendix**

I liked the idea of the paper but would like to emphasize that negative results on a particular dataset should be included in the main paper instead of the appendix. In this paper, the negative results on FB15K-237 were reported only in the appendix and no experiment results on FB15K-237 were reported in the main paper. FB15K-237 was listed in Table 1, "Statistics of various datasets used in experiments", therefore excluding it from the result session makes the results incomplete, and many interesting discussion and analysis that should have been done and presented were omitted as a result. It also makes the last sentence in the abstract "MINERVA obtains state-of-the-art results on seven KB datasets, significantly outperforming prior methods" a severe overclaim.

More importantly, FB15K-237 is a challenging knowledge graph modeling testbed curated by (Toutanova and Chen, 2015), which is more at-scale and more wildly studied than a few of the datasets the paper used (COUNTRIES, KINSHIP, UMLS). Inferior performance on this dataset could indicate serious flaws of the proposed model when applied on open-domain KGs.

Many readers pay attention to the appendix only when necessary. The paper should include pointers to the appendix for contradictory results to this degree. Leaving the results in appendix without making additional claims creates a bias for reviewers and other readers, and should be strictly discouraged.

---

> ### Author Response · Authors · 2017-11-26
> **Re:**
>
> We agree that reporting negative results on a particular dataset is important and that's why we chose to include them in our submission.
> i) There has been a high variance of scores reported for FB15k-237 among many recent papers (ranging from low 40 hits@10 to ~53 hits@10). In fact, the results we currently get is comparable to state-of-the-art results reported in some recent papers. However, instead, we chose to compare to a very high score which our in-house implementation achieved.
> ii) We politely disagree that FB15k-237 is a great dataset for comparing query answering performance. The presence of many 1-to-Many query relations, low clustering coefficient (Holland & Leinhardt, 1971; Watts & Strogatz, 1998) and low occurrence of various path types makes it less amenable for path based models and less interesting for query answering. We will update the next version of the paper with more detailed analysis.
> iii) The 7 datasets where MINERVA achieves excellent results are i) Countries ii) UMLS  iii) Kinship iv) WN18RR v) NELL-995 vi) WikiMovies vii) Grid World (important experiment since we test how MINERVA works for long paths)
>
> Lastly, we would like to emphasize that the community should be more welcoming to negative results and analysis. Comments such as this only discourages young researchers to not report negative results leading to bad science.
>
> Paul W Holland and Samuel Leinhardt. Transitivity in structural models of small groups. Comparative Group Studies, 1971.
> Duncan J Watts and Steven H Strogatz. Collective dynamics of small-worldnetworks. nature, 1998.

---

> > ### Public Comment · (anonymous) · 2017-11-26
> > **The community's perspective to negative results**
> >
> > Quoting a paragraph of the author response:
> >
> > "Lastly, we would like to emphasize that the community should be more welcoming to negative results and analysis. Comments such as this only discourages young researchers to not report negative results leading to bad science."
> >
> > It is a great point to be brought up that *every* researcher should not be obstructed by negative results and the community should be more welcome to them. However, I think this is the point of the main comment, not vice versa. The comment made it clear that it is demanding negative results to be included in the main body of a paper and interesting discussion and analysis to be done and presented, not demanding to "not submit a paper with negative results". It is advocating for legit presentation of negative results instead of presenting them in supplementary, separated from positive ones.

---

> ### Author Response · Authors · 2017-12-13
> **Results moved to the main body of the paper**
>
> We have moved the results to the experimental section of the paper (Sec 4.1) and also provided detailed analysis regarding the results. Thanks for the suggestion.

---

### Public Comment · (anonymous) · 2017-12-08
**Not fair when comparing with other previous model not using relation path**

Your model exploits information about relation paths, so you should carefully denote the experimental results which previous models use the relation path information to avoid a confusion. See an example from [1]. It's not fair when you compare your model with other models not using the relation path information.

[1] Knowledge Base Completion: Baselines Strike Back. Rudolf Kadlec and Ondrej Bajgar and Jan Kleindienst

---

> ### Author Response · Authors · 2018-01-05
> **Paper updated with additional experiments**
>
> We have updated the paper with a new experiment which explicitly compares with a neural multi-hop (path) model which is designed to work in the query answering setting (when the target entity is unknown). Please refer to sec 4.2 of the paper. Thanks!

---

### Public Comment · (anonymous) · 2017-12-09
**Evaluation is not convinced**

While I like the idea of exploiting relation path information in the KB to improve link predictions, I am not convinced by the experimental results: why are MR or MRR scores not reported in the paper? It is abnormal with respect to KB completion evaluation.

It should definitely be straightforward to compute MR and MRR scores while you already have a model optimized on Hits@10.

I suspect the  proposed model could not produce competitive MR or MRR results against the baselines? So, experiments cannot prove the  proposed model to be useful.

---

> ### Author Response · Authors · 2017-12-21
> **Re:**
>
> Thank you for the comment. We have updated the paper with mean reciprocal rank (MRR) and median rank (MR) and  compared with baselines whenever they were reported. For Countries and DeepPath, the evaluation metric is AUC-PR and MAP respectively and they are equivalent to MRR when there is one correct answer [1,2]. We outperform baselines on these metrics and hence the results are in agreement with Hits@k and accuracy,
>
> [1]https://stats.stackexchange.com/questions/127041/mean-average-precision-vs-mean-reciprocal-rank
> [2]https://stats.stackexchange.com/questions/157012/area-under-precision-recall-curve-auc-of-pr-curve-and-average-precision-ap

---

### Public Comment · (anonymous) · 2017-12-21
**MR is unbelievable. Reviewers should aware this.**

I do not believe your MR results on WN18 and FB15k-237. There are two main reasons:

+ You compare your model with ConvE, Complex and DistMult. MR of ConvE, Complex and DistMult are 7323, 5261 and 5110 on WN18RR and 330, 248 and 254 on FB15-237 (shown in [1]). You absolutely know this but you hide these results. Other similar results are also reported in [2]. But your MR results of 10 on WN18 and 18 on FB15k-237 is unbelievable because there is a too big difference between results.

+ Lower MR shows better performance. You did not tell the reviewers that WN18RR is a subset of WN18 and FB15k-237 is a subset of FB15k. As mentioned by [1], the test sets of WN18 and FB15k contain mostly reversed triples that are present in the training set. So WN18RR and FB15k-237 are created for the link prediction task much more realistic and challenging. So, it's natural that the MR values on WN18RR and FB15k-237 are higher than those on WN18 and FB15k (see MR results on WN18 and FB15k in [2,3,4]). But your MR values of 10 on WN18RR and 18 on FB15k-237 are significantly lower than state-of-the-art values on WN18 and FB15k.

Two above reason make me that I also do not believe in your MRR and Hits@10 results.

[1] Convolutional 2D Knowledge Graph Embeddings
[2] Revisiting Simple Neural Networks for Learning Representations of Knowledge Graphs
[3] Knowledge Base Completion: Baselines Strike Back
[4] An overview of embedding models of entities and relationships for knowledge base completion

---

> ### Public Comment · ~Sachin_Rajoria2 · 2017-12-21
> **Be Constructive**
>
> I am sorry that you do not believe in the results but I don't understand your attitude.
> Can you please explain why you think it is unbelievable??? If hits@10 is larger than 0.5
> it means the median rank (MR) will upper bounded by 10.  Also most KB completion papers
> don't use MR.

---

> > ### Public Comment · (anonymous) · 2017-12-22
> > **I am so sorry. However, the evaluation protocol is not convinced**
> >
> > It's my mistake when not seeing MR (median rank) carefully. I thought it's mean rank. I am so sorry for that you do not follow the standard metric (mean rank MR) when doing the knowledge base completion task.
> >
> > I agree with the aspect of question-answer. However, when you are trying to adapt your model to a new task, i.e., the knowledge base completion task that you are not familiar, your evaluation protocol is still not convinced.
> >
> > 1.	You should report three metrics:Hits@k, MRR and Mean Rank MR. Because there are standard metrics for the task, then adding the median rank results if you want.
> >
> > 2.	A trained model is obtained after tuning the hyper-parameters on the validation set with respect to just one single metric (e.g., Hits@k or MRR). After that, the trained model is used to report Hits@k, MRR and mean rank MR on the test set. For each dataset, Hits@k, MRR and MR results are come from the *same* trained model.
> >
> > 3. 	However, you always say "training a new model" to report addition results. It means that you tune hyper-parameters on the validation set with respect to each metric, i.e., for each dataset, *your results are come from different trained models*. This is not right to evaluate the knowledge base completion task and not fair when comparing with other previous models.
> >
> > So, after all, even you add new results, your results are still not convinced to me.

---

> > > ### Author Response · Authors · 2017-12-22
> > > **re:**
> > >
> > > It is incredible that the commenter continues to be so rude and misleading (should OpenReview have a moderating system?), and continues to frame this interaction as an attempt to convince *them* rather than to correct the constant series of willful misinterpretations and falsehoods that they manage to state about our work in every single interaction, in the hope that they do not mislead others. If they are unconvinced, they are free to not use our code or build on the work.
> > >
> > > We use the same model, per dataset, for each evaluation metric.  Our previous comment about training a new DistMult (which it could be noted is not our model, but in fact a baseline against which we are competing) WordNet model to report median rank was because we did not have our own model for WordNet DistMult to report median rank, and were using numbers from prior work for the other metrics. While we are touched by your concern that we may be attempting to make our baselines too powerful, we are following the standard procedure.
> > >
> > > Contrary to the assertion, as we provided copious evidence for in the last response, "mean rank" is not a particularly standard metric for the task (with which the authors are extremely familiar), is unreported by many papers, and has problems of its own in over-weighting outliers and being heavily data-size dependent. This is in addition to its lack of meaning for our model. In our last comment, we described the situation in some detail, not for the hostile commenter's benefit, but for any passers by who could be mislead by the litany of false statements they have made at every step of this regrettable interaction.
> > >
> > > At this point we urge the commenter to work on their own research rather than discuss ours, as they seem unable to do the latter without stating untruths about the work, or impugning the personal ethics and/or common sense of the authors, without having so much as properly read our evaluation protocol.
> > >
> > > Regardless, no further comments from this person will receive a response, as bad faith has been more than adequately demonstrated.

---

> ### Author Response · Authors · 2017-12-21
> **re:**
>
> We thank the commenter for an unintentional suggestion to add the *median* rank results for WN18RR DistMult, which we did not have since it requires training a new model. Since, as reported in the same paper the commenter cites [1], DistMult on FB15-237 has reproducibility issues (which we also mention in a footnote), we trained our own model to report those numbers, and were easily able to evaluate *median* rank for comparison.
>
> As for the suggestion that we falsified results by hiding numbers that we do indeed “absolutely know,” we refute these categorically, find them to be in extremely poor taste, and generally, think one should very carefully read the results section of a paper to be 100% sure before making such a serious accusation.
>
> Because our model is for query-answering rather than pure triplet-scoring, we perform a beam search at inference time, which we clearly define at the beginning of 4.1. Accordingly, *mean* rank is not a good fine-grained measurement of model quality --- any answers that fall off the beam would have to pessimistically be put all the way at the bottom of the results list. We do not evaluate mean rank or include mean rank results for previous models, of which we are indeed aware, because the metric provides little insight for our model.
>
> MRR does not suffer from this problem, as low-ranked answers past a certain point contribute essentially 0, and *median* rank does not have this problem unless our model was so bad that it was unable to get the right answer in the beam (40) even half the time. So, we decided to report the median rank, as we very clearly stated, and defined the acronym of, at the beginning of 4.1. Since FB15k-237 is a dataset of broad interest and we had an implementation, we included this number for DistMult as well for comparison.
>
> We are unaware that “MR” is an industry-standard notation for a “mean rank” evaluation, or indeed that it is even a particularly common evaluation at all outside of a few recent papers that you cite. We do not see it, for example, in this large list of metrics for information retrieval [3], or many books or course notes on IR (e.g. the Stanford IR book [4]). For example, the seminal Trans-E work [2] simply refers to it as “Mean Rank”, and the original paper creating FB15k-237 [5], does not even report it. However, we will change the name to “MedR” to avoid confusion. It never crossed our minds that any such confusion would result from using this abbreviation to denote *median*, since as you note, these numbers are 2 decimal orders of magnitude lower than reported in previous literature, while the rest of the metrics are middle-of-the-pack. In fact, we are reporting that DistMult beat us by 8 to 18 on this result. The same metric on which you accuse us of leaving out known, reported numbers for the other models --- to put it in plain English, of falsifying!
>
> Regarding the comment “You did not tell the reviewers that WN18RR is a subset of WN18 and FB15k-237 is a subset of FB15k”: We actually mention specifically in Section 3, Data, that WN18RR is a subset of the original WORDNET18 dataset, and cite relevant papers for the rest, though we don’t see why that has any bearing on anything, let alone meriting a conspiratorial accusation like “you did not tell the reviewers.”
>
> In final, we will train a model and add *median* rank results for WN18RR DistMult using our implementation, for completeness, change the name to “MedR” to avoid any confusion, and perhaps add a note about the unsuitability of mean rank for evaluating a query-answering model on binary-labeled triplets.
>
> Had those been suggestions made in the comment, it would have been a helpful one. In the future, I think the commenter would be well-served by carefully reading a paper before accusing other scientists of falsification and suggested fabrication.
>
> [1] Convolutional 2D Knowledge Graph Embeddings
> [2] Translating Embeddings for Modeling Multi-relational Data
> [3] https://en.wikipedia.org/wiki/Evaluation_measures_(information_retrieval)
> [4] https://nlp.stanford.edu/IR-book/pdf/08eval.pdf
> [5] Representing Text for Joint Embedding of Text and Knowledge Bases

---

### Decision · Program_Chairs · 2018-01-29
**ICLR 2018 Conference Acceptance Decision**

**Decision:**

Accept (Poster)

**Comment:**

Good contribution. There was a (heated) debate over this paper but the authors stayed calm and patiently addressed all comments and supplied additional evaluations, etc.